# Neuromorphic Sensor Based on Force-Sensing Resistors

**DOI:** 10.3390/biomimetics9060326

**Published:** 2024-05-29

**Authors:** Alexandru Barleanu, Mircea Hulea

**Affiliations:** Department of Computer Engineering, Gheorghe Asachi Technical University of Iasi, 700050 Iasi, Romania

**Keywords:** neuromorphic sensors, force-sensing resistors, spiking neurons, shape memory alloy, anthropomorphic finger control

## Abstract

This work introduces a neuromorphic sensor (NS) based on force-sensing resistors (FSR) and spiking neurons for robotic systems. The proposed sensor integrates the FSR in the schematic of the spiking neuron in order to make the sensor generate spikes with a frequency that depends on the applied force. The performance of the proposed sensor is evaluated in the control of a SMA-actuated robotic finger by monitoring the force during a steady state when the finger pushes on a tweezer. For comparison purposes, we performed a similar evaluation when the SNN received input from a widely used compression load cell (CLC). The results show that the proposed FSR-based neuromorphic sensor has very good sensitivity to low forces and the function between the spiking rate and the applied force is continuous, with good variation range. However, when compared to the CLC, the response of the NS follows a logarithmic-like function with improved sensitivity for small forces. In addition, the power consumption of NS is 128 µW that is 270 times lower than that of the CLC which needs 3.5 mW to operate. These characteristics make the neuromorphic sensor with FSR suitable for bioinspired control of humanoid robotics, representing a low-power and low-cost alternative to the widely used sensors.

## 1. Introduction

The spiking neural networks represent the third generation of neural networks which introduces time in information processing and learning. The SNNs are based on spiking neurons that model rigorously the physiology of the neural cells in the brain. The hardware implementation of the SNN benefits from fast response, low power consumption, and a very good signal-to-noise ratio [1], which makes SNN a good candidate for the implementation of control units of robotic systems. In robotic systems, motion is of high importance because it provides the robots with the ability to interact mechanically with the environment. Typically, the motion is performed by electric, pneumatic, or hydraulic actuators, while the control of motion is based on the feedback from neuromorphic sensors. This type of sensor includes a sensing component and additional electronic circuits that convert the measured quantity into trains of impulses with parameter-dependent frequency [2]. The advantage of the sensors with spiking output over the sensors that generate continuous signals is the direct interaction with the spiking neural networks. In addition, by encoding the information with short pulses (spikes), the power consumption is reduced significantly due to the passive operation of the sensor between spikes. There are several materials to implement the sensing element of a pressure sensor, such as ferromagnetic powder in a resin [3], or piezo-resistive film, which also detects the contact position using a conductive film [4]. Moreover, multipoint sensing was implemented with Euler–Bernoulli bending rods that model the sensing mechanism of whiskers in living organisms [5]. Other types of tactile sensors with spiking output are built using one organic transistor for sensing, and another transistor for pulse generation [6]. An advanced study focused on the implementation of a mechanoreceptor with spiking output that responds to pressure and vibration modeling natural tactile sensing. This system is able to communicate with the nerves in living organisms and to discriminate between textures of the surfaces [7].

The sensing components of such sensors can use piezo-electric field effect transistors FET that convert the physical measure into electrical current [8]. In neuromorphic sensors, this current is used by additional analog hardware to generate spikes with parameter-dependent frequency [9]. Another method to convert physical measures into spikes is to use threshold-switching memristors fabricated using NbOx that have resistive switching characteristics for low currents [10]. Therefore, the recently developed memristors mimic the behavior of the biological synapses by generation of pulses with different frequencies [11] and adaptability [12]. Recent research shows that memristors-based technology can be used to develop advanced sensors of high biological plausibility such as mechanoreceptors [13] or nociceptors [14]. The spiking rate generated by these devices depends on the input produced by the mechanical sensors that produce the energy consumed by the receptors [13].

The sensing components mentioned above are expensive or have limited availability. In contrast, a force-sensing resistor (FSR) is cheap and easy to use, making this type of component suitable for a wide range of applications.

### 1.1. Force-Sensing Resistors

FSR uses polymer thick film technology to convert the force into resistance that varies in the opposite direction to the applied force. The FSR is built of two layers of flexible polyester film substrate, each with a layer of conductive silver ink that constitutes the electrodes. Between the two electrodes, there is a compressible polymer that responds to the applied force by increasing the conductance [15]. The alignment of electrodes relative to the composite material influences significantly the response of the sensor [16]. This implies that the transverse position of the electrodes offers better sensitivity than the sandwich structure [17]. With no applied force, the sensor behaves as an open switch due to infinite resistance, and when the sensor is pressed, the drop in resistance is significant, even if the equivalent mass is below 100 g. Note that the resistance reduces exponentially with the mass that can reach several kilograms. With proper design of the sensing component, the repeatability of the measurements for one sensor is below ±2%, showing good reliability of the FSR [15]. In order to improve the FSR response, several calibration techniques were developed for the correction of errors, such as hysteresis [18] and creep [19].

The applications of FSR start from the implementation of wearable sensors [20], including gaming [21], to prostheses for humans in biomedical engineering [22], and for monitoring cardiorespiratory parameters during sleep [23]. Considering the high sensitivity of FSR to small forces, these sensors are suitable for interaction with human hands in musical instruments [24]. In another application, several FSRs that are distributed on the human soles of feet are used by a NN-based application to determine the center of weight while standing or walking [25].

### 1.2. Compression Load Cells

A CLC converts the applied force into voltage based on a strain gauge (SG), which represents the measuring component of the load cell. The strain gauge is a small grid consisting of wires made of an alloy of copper and nickel that is sensitive to the deformation of the load cell surface. This changes the resistance of the SG that can be converted into voltage with an electronic circuit that is based on the Wheatstone bridge. This component connects multiple SGs and detects deformation as a change in electrical signal [26]. The CLC benefits from high linearity and repeatability of the response even at extremely high load capacities making this type of sensor suitable for weight measurement. In robotics, CLCs are useful to implement force sensing in grippers during grasping and holding, as well as to evaluate the compression force on the robot’s feet during walking [27]. However, despite its accuracy and reliability, the compression load cells typically come at a higher cost and lower sensitivity to small forces than the FSR.

### 1.3. SMA Actuators

Among the electric actuators, the shape memory alloy (SMA) represents an elegant alternative to the widely used motors because of their small size, low weight, and significant force–weight ratio [28]. The SMA is a metallic material that changes its state when heated from the martensite phase when cold to the austenite phase, which represents the memorized shape. During the austenite phase, which is determined by higher temperature, the actuator shortens its length with a significant force. For example, an actuator made of nitinol with a 0.006 inch diameter can pull up a mass of 320 g according to the technical data. Also, the lifetime of SMA is of the order of 10^6^ actuation cycles, implying that actuators have significant reliability. However, there are some disadvantages such as the nonlinearity due to the difference between the contraction and relaxation temperatures, respectively. The environmental temperature influences several parameters of SMA, such as the initial length, the contraction speed, and the relaxation time when no auxiliary cooling method is used. In addition, the change in length during contraction is about 4%, which increases the length of necessary actuators for long displacements.

Despite these disadvantages, the SMA benefits from the same actuation principle and is silent in operation as the natural muscles are suitable for the actuation of the bioinspired systems. In addition, the spiking neurons are able to control directly the contraction of SMA actuators in a biomimetic manner by the firing rate [29,30].

### 1.4. The Goal of the Current Research

In this work, we proposed a neuromorphic sensor based on an electronic spiking neuron that includes the FSR in its schematic. Compared with the existing neuromorphic force sensors, the proposed NS_FSR_ integrates a cheaper sensing component that simplifies the structure of the CLC-based sensors [29] and reduces the cost of FET-based ones [9]. In addition, the spiking neuron used by NS_FSR_ has similar operation principles as the memristors, such as action potential, threshold-driven spiking, refractory period, and strength-modulated frequency response. However, these features are obtained using electronic components that are significantly cheaper and more affordable than the NbOx that is fabricated using complex techniques. Moreover, being based on discrete components, the spiking neuron can be easily integrated into an analog chip to reduce its dimensions. Compared to the CLC-based sensors that use the same model of spiking neurons, the NS_FSR_ benefits from a simplified electronic circuit due to the integration of FSR in the neuron schematic.

Recent research showed that FSR is a cheap component that is suitable for measuring the grasping force of both robotic [31] and human fingers [20,32]. Starting from these findings, the proposed NS_FSR_ is validated in a robotic application where SNN controls the force of an anthropomorphic finger actuated by SMA. The behavior of the finger is evaluated when the FSR-based sensor and the CLC are used independently to sense the force on the finger’s tip.

The remainder of the paper is organized as follows: Section 2 details the schematics of the SOMA with FSR and the structure of the SNN for evaluation of the NS_FSR_ and CLC in robotic finger control, together with the experimental setup. The results are presented in Section 3, followed by the conclusions that focus on the advantages of the proposed sensor and future work.

## 2. Materials and Methods

The proposed sensor is based on an electronic spiking neuron of high biological plausibility [30] because it obtained good results for the bioinspired control of SMA-actuated anthropomorphic fingers using the feedback from sensors [29,30]. In addition, taking into account that the activation rate of the SN depends on the value of the input resistor $R_{E}$, the neuromorphic sensor is obtained by connecting FSR instead of $R_{E}$. Thus, the neuron model is selected for implementation of the NS_FSR_ due to the very simple adaptation of the schematic, and the demonstrated performance in robotic control.

### 2.1. Electronic Neuron

This neuron model includes one input module denoted SOMA for integration of the incoming pulses and activates the synapses (SYNs) when the activation threshold is detected. The SYNs store the synaptic weights and generate excitatory or inhibitory spikes whose intensity depends on the weights. In addition, the SYNs model the Hebbian learning mechanisms that make the neuron respond to the temporal coincidence of incoming stimuli.

### 2.2. The Neuromorphic Sensor

Figure 1 presents the schematic of the SOMA that includes the capacitor $C_{M}$ for integration of the incoming spikes, and the transistors $T_{TH}$, $T_{ACT}$ for detection of activation threshold $V_{TH}$ and triggering the synapses’ activity. When $V_{TH}$ is reached, the SOMA is active for a fixed period of time $t_{ACT}$ when the synapses generate a spike. An important characteristic of the neuron represents the possibility of activating the SOMA by a continuous potential $V_{E}$ that determines its activation rate $f_{N}$.

Starting from this characteristic of the electronic neuron, the neuromorphic sensor uses the FSR instead of $R_{E}$ and a fixed potential $V_{FSR}$ to activate the neuron. The value of $V_{FSR}$ is chosen empirically to maintain the high sensitivity of the sensor, and to increase the range of the spiking rates generated by the SOMA. In order to avoid the use of an additional power supply, $V_{FSR}$ is adapted to the VDD through the auxiliary circuit AUX, which is highlighted in blue in Figure 1. In this setup, the modified SOMA drives the synapses that generate spikes whose frequency depends on the applied force on the FSR.

### 2.3. The Structure of the SNN

We tested the neuromorphic sensor NS_FSR_ in a control application where the force of an anthropomorphic finger is regulated by a SNN that has the main structural characteristics of the biological motor neural areas presented by the neuroscientific research [33]. The SNN includes excitatory neurons that are driven by a command, and inhibitory neurons that regulate the activity of motor neurons based on the sensor’s input. According to Figure 2a, the neurons $E_{1,4}$ in the input layer activate the motor neurons $M_{1,2}$ in the output layer that drives the SMA actuator. The feedback from the sensor NS_FSR_ is received through the inhibitory synapses $IS_{1,2}$ that regulates the activity of $M_{1,2}$. For comparison with CLC, we replaced the NS_FSR_ with this load cell that activates an inhibitory neuron using the circuit POT as in Figure 2b.

The motor neurons stimulate through the SMA driver the actuators whose contraction force is determined directly by the neurons’ spiking rate. Note that the frequency of the motor neurons depends on the resultant activity of NS_FSR_ and $E_{1,4}$, of which the firing rate $f_{E}$ depends on $V_{E}$. Therefore, considering that the firing rate of motor neurons increases with the frequency of excitatory neurons, the force of the SMA actuator can be adjusted using $V_{E}.$

### 2.4. Experimental Setup

We validated the proposed sensor using an anthropomorphic robotic finger that is flexed by a SMA actuator and has two force sensors on the finger’s tip as in Figure 3. The actuator is implemented with 0.006 inch wires type Flexinol LT (Dynalloy, Inc., Irvine, CA, USA) that can reach up to 4% stroke and 321 g force when a maximum of 410 mA heats the wire. For the reported experiments, the length of the actuator is 115 cm, which is powered at $V_{CC}=24\;V.$ We compared the performances of a compression load cell type FS2050 (TE Connectivity, Berwyn, PA, USA), and the force-sensing resistor type FSR03CE (Ohmite, Warrenville, IL, USA). In order to reduce the influence of the contact area on the FSR response and other surface-related errors [16,22], we placed a metal disk with a diameter of 20 mm in front of the sensor. The system is evaluated by measuring the output of the CLC and the neuron’s frequency during steady state when the finger pushes on an elastic tweezer. Note that the CLC is below the FSR in order to evaluate the behaviors of the finger with the two sensors in the same conditions. During the experiments, the FSR-based sensor and CLC are not connected to the input of the SNN, simultaneously implying that the use of two sensors for feedback is exclusive. However, the information about the finger’s force is obtained by reading the analog output of the CLC, even if the NS_FSR_ feeds the SNN. In order to deepen the comparison between the two sensors, we determined the energy consumptions for the NS_FSR_ and, respectively, of the CLC by measuring using a digital multimeter type SDM3065X (SIGLENT Technologies Germany, Augsburg, Germany) the electrical current that is used by the two circuits.

Figure 4 presents the diagram of the experimental setup that is based on the SNN structure presented in Figure 2. Note the SNN connected to the SMA driver and the two sensors that are connected to the finger’s tip. 

The CLC is connected through a potentiometer POT to one inhibitory neuron included in the inhibitory area of the SNN. The proposed neuromorphic sensor includes the FSR that is connected to the SOMA to drive the corresponding inhibitory synapses that are included by the inhibitory area.

## 3. Results

The experimental work is split into two phases that are focused on the evaluation of the NS_FSR_ and CLC response outside the system, and on the validation of NS_FSR_ in a SNN-based system for robotic finger control.

### 3.1. Sensors Response with the Load Mass

During the first experimental phase, we determined the variation in the firing rate $f_{E}$ of the SOMA with the input voltage $V_{E}$, followed by an evaluation of the response of CLC and NS_FSR_ with the load mass $m_{L}$. As presented in Figure 5a, the activation rate $f_{E}$ of the SOMA varies linearly with $V_{E}$ when $R_{E}$ is fixed, and, similarly, the CLC output increases linearly with $m_{L}$ (see Figure 5b). This behavior implies that the spiking rate of the inhibitory neuron is proportional to the load mass when the CLC is used as a force sensor.

The linearity of SOMA’s response shown in Figure 5a is evaluated theoretically by analyzing the variation of $f_{E}$ with $V_{E}$ using the schematic presented in Figure 1. The frequency $f_{E}=1/\Delta t_{CH}$ where $\Delta t_{CH}$ is the charging period of the capacitor $C_{M}$ when the input potential $V_{M}$ varies by $\Delta V_{M}=V_{BE}-V_{R}$. The potentials $V_{BE}$ and $V_{R}$ are the emitter-base voltage of the transistor $T_{TH}$, and, respectively, the minimum value of the neuron’s input after activation. Using Ohm’s law, we can write the variation of the charging current during $\Delta t_{CH}$ as follows:(1)$$\Delta I_{CH}=\Delta V_{M}/R_{E}$$
showing that $\Delta I_{CH}$ does not depend on $V_{E}$. In addition, when $R_{E}\gg$ $\Delta V_{M}$, the variation of the current is insignificant, implying that $C_{M}$ is charged by an almost constant current $I_{CH}$, which depends on $V_{E}$. In addition, we can determine the influence of $V_{E}$ on the charging interval as follows:(2)$$\Delta t_{CH}=-R_{E}\cdot C_{M}\cdot \ln\left(1-\frac{\Delta V_{M}}{V_{E}}\right)$$

Using the parameters given in Figure 1, the derivative ${\Delta t_{CH}}^{\prime }$ of expression (2) is insignificant. For example, if we denote by $dt_{3V}$ and $dt_{10V},$ the values of ${\Delta t_{CH}}^{\prime }$ when $V_{E}$ is 3 V and 10 V, respectively, with step $\Delta V_{E}=0.25\;V$, the ratio $r=dt_{3V}$/$dt_{10V}\approx 1.0014$. This implies that ${\Delta t_{CH}}^{\prime }$ is almost constant, implying that the activation rate $f_{E}$ increases almost linearly with $V_{E}$, as demonstrated experimentally in Figure 5a. Note that according to Ohm’s law, the neuron response has similar linearity with the variation of $R_{E}$. This implies that the nonlinearity of the NS_FSR_ response presented in the sequel is determined only by the FSR that replaces $R_{E}$.

For evaluation of the CLC’s response presented in Figure 5b, we used 14 disks of 3.5 g that are stacked sequentially above the sensor. Considering that the minimum mass that activates the inhibitory neuron is $m_{\min}=11\;g$, we added an offset mass of 7.5 g. The CLC output was read for each additional disk when the mass varied between 11 to 56 g with a resolution of 3.5 g. In order to assess the repeatability of the results we performed this set of measurements 5 times and plotted the values with different colors in Figure 5b. The output of the CLC is proportional with $m_{L}$ and variation between the plots is not significant, showing good repeatability. Similarly, for evaluation of the FSR-based neuromorphic sensor NS_FSR_, we performed 5 sets of measurements when the load mass ranges between 11 and 56 g (see Figure 6). For this sensor, we measured the output frequency $f_{NS}$ of the neuron that is directly related to the FSR value and consequently to the applied force.

Note that the spiking rate generated by the neuromorphic sensor increases with $m_{L}$ following a logarithmic-like function mainly above 25 g for all sets of measurements that are plotted with different colors. This behavior is a consequence of the nonlinear function that describes the FSR response [15]. In addition, the difference between the plots is higher than that in Figure 5b, showing that the repeatability of the results with a FSR-based sensor is lower. For this experimental setup, the maximum value of $m_{L}$ is in the range supported by SMA actuated robotic finger where the variation of the FSR response is higher. However, the NS_FSR_ supports significantly stronger forces but the discrimination power of the sensor is significantly lower (i.e., $f_{NS}=610\;\mathrm{Hz}$ for $m_{L}=0.5\;\mathrm{kg}$ and $f_{NS}=615\;\mathrm{Hz}$ for $m_{L}=1\;\mathrm{kg}$).

### 3.2. The Control of the Robotic Finger

For the second experimental phase, we implemented a robotic finger that is actuated by SMA and includes both force sensors on the distal phalange. In this setup, we evaluated the ability of this robotic finger to push and hold one tip of an elastic tweezer, while the other tip is blocked. The contraction force of the actuator is adjusted by changing the input voltage $V_{E}$ that activates the excitatory neurons (EN) at a constant frequency. For a clear view of the SNN activity, Figure 7 and Figure 8 show oscilloscope recordings with the pulses generated by the SOMAs during neurons’ activity. In these diagrams, the signals illustrate the activity of motor neurons MN (blue signal) that integrates the activity of EN (green signal) and of the NS_FSR_ (magenta signal).

In Figure 8, the force of the SMA actuator varies due to the intermittent activity of the motor neurons, implying that for lower frequencies of the excitatory neurons, the regulatory performance of the SNN decreases. In contrast, for higher frequencies of the EN, the variation of the finger’s force during the regulatory regime is insignificant.

Figure 9a presents the variation of the CLC output with the frequency of the excitatory neurons when this CLC drives the inhibitory neurons as in Figure 2b.

The frequency $f_{E}$ of EN is set to several values between 350 and 550 Hz with a resolution of 25 Hz. The limits of the variation interval are chosen empirically to reduce the oscillation of the finger force for the lowest $f_{E}$ and to reach maximum contraction force for higher $f_{E}$. Note that above $f_{E}=550\;\mathrm{Hz}$, there is no significant motion of the finger when it pushes on the tweezers. Similarly, the variation of $V_{F}$ with $f_{E}$ is given in Figure 9b when the proposed NS_FSR_ regulates the activity of the motor neurons. Taking into account the linearity of the load cell’s response, in this case, the CLC is also used to measure the force of the finger on the tweezer’s tip. During all the measurements, the synapses are potentiated to the maximum weights that are determined by the activation of the input neurons by at least 5 s.

At low forces, the variation of the FSR resistance is significant, implying that the sensitivity of the NS_FSR_ to low forces is high. Indeed, a mass of only 4 g with a small contact area placed directly on the FSR surface without the 20 mm disk (see Figure 3) triggers the activity of NS_FSR_. In contrast, the same mass changes the CLC output by only 8 mV, which is insignificant related to the variation range [1,4] V of the sensor’s output. Thus, despite the linearity of the CLC’s response, the sensitivity to small forces is lower than that of the FSR-based sensor.

Considering that power consumption is another important parameter for neuromorphic systems, we measured the electrical current used by the sensors. For the maximum force in the experiments, the obtained values are ~7 mA at 5 V for the CLC and ~80 µA at 1.6 V for the NS_FSR_. For no load, there is no significant change in the current for CLC, but it reduces to 20 µA for the NS_FSR_. This implies that the power consumption of the neuromorphic sensor is more than 270 times lower than that of the CLC and reduces for smaller loads.

## 4. Conclusions

In this work, we proposed a neuromorphic sensor based on an electronic spiking neuron that is easily adapted to include a force-sensing resistor. The frequency of the output spikes that are generated by the sensor varies with the FSR resistance and consequently with the applied force. The validation of the proposed sensor was performed in two experimental phases: During the first one we measured directly the spiking frequency when several load masses with known values were placed on the sensor, and during the second phase we integrated the sensor in a SMA actuated robotic finger and measured the force during steady state when the finger pushes on an elastic tweezers.

The evaluation of the sensors’ response shows that the output of the CLC has a high degree of proportionality with the load mass and, consequently, with the applied force, implying that the output of the neuron driven by the CLC is linear. On the other hand, the variation of the output frequency of NS_FSR_ with the load mass follows a logarithmic-like function that is determined by the nonlinearity of the force resistance function of the FSR. The results of the second experimental phase show that the force of the robotic finger is constant for the rates of excitatory neurons that are above 400 Hz. This implies that despite the lower linearity of NS_FSR_ compared with the CLC, this FSR-based sensor is suitable to discriminate a wide range of applied forces when proper nonlinear control is implemented. In addition, the power consumption of the NS_FSR_ is 0.128 µW, which is significantly lower than the 3.5 mW needed by CLC. This advantage makes the proposed sensor useful not only for the implementation of bioinspired control systems in robotics, but also for low-power electronics based on neuromorphic hardware.

For future work, we intend to reduce the power consumption of the neuromorphic sensor, which can be increased even further by redesigning the neurons’ schematic for FET-based technology and implementing an integrated chip (IC). The first task to fulfill this goal is to reduce the values of capacitors in the pF range that reduces significantly the physical dimensions of the IC [34]. A technology that we can use to make the IC implementation feasible is AMS 0.35 µ SiGe-BiCMOS S35D4M5/CMOS-RF C35B4M3 4M/4P Thick MET4—MIM. In addition, we will improve the sensor to include a network of spiking neurons that can compensate for the nonlinearity of the FSR.

## Figures and Tables

**Figure 1 biomimetics-09-00326-f001:**
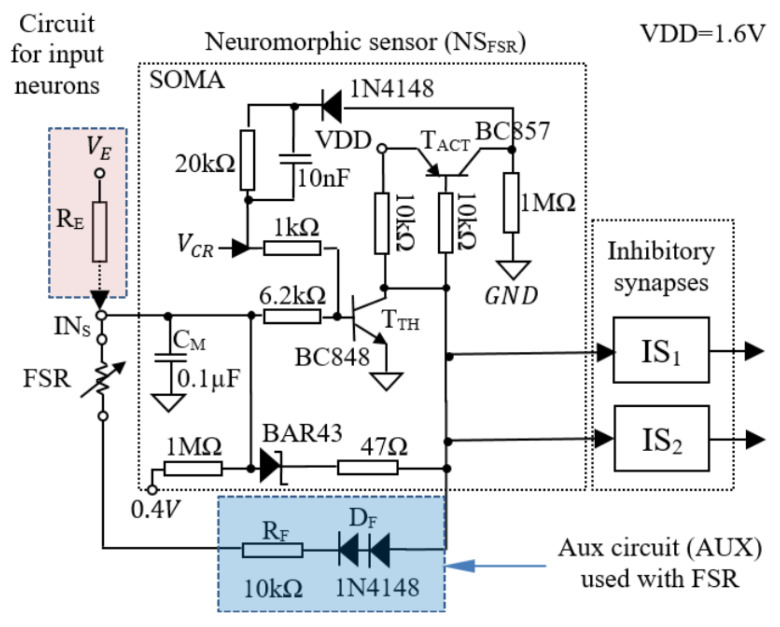
Schematic of the neuromorphic sensor based on FSR including one SOMA and two inhibitory synapses IS_1_, IS_2_, as well as the additional components for adjusting the generated frequency range.

**Figure 2 biomimetics-09-00326-f002:**
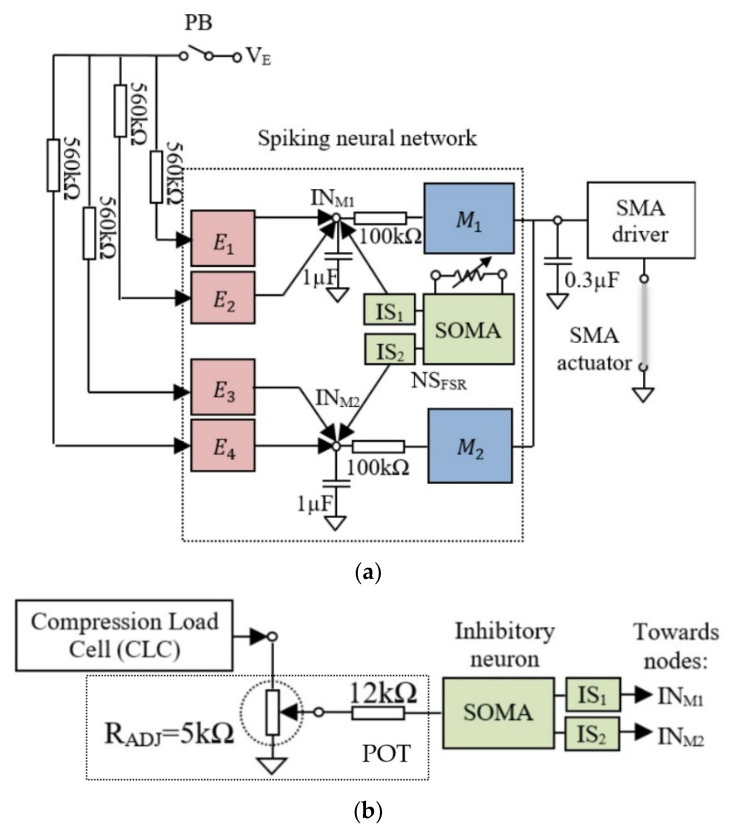
(**a**) The structure of the SNN used for SMA control using the FSR-based neuromorphic sensor (NS_FSR_); (**b**) the circuit that replaces the NS_FSR_ when the compression load cell is used to sense force.

**Figure 3 biomimetics-09-00326-f003:**
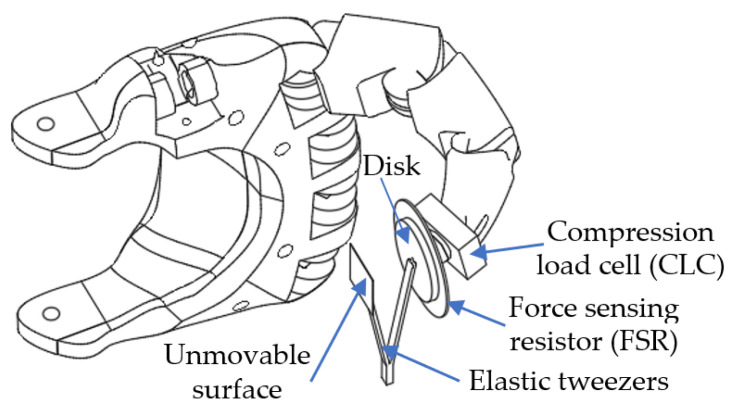
The structure of the finger with the FSR and CLC on the finger’s tip.

**Figure 4 biomimetics-09-00326-f004:**
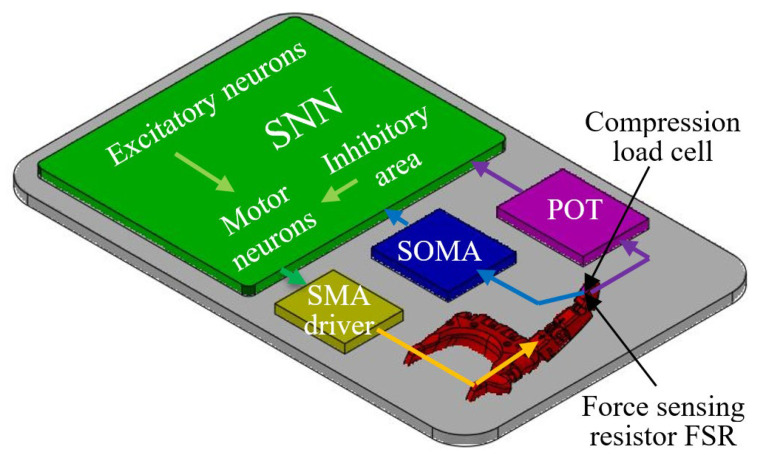
The diagram of the experimental setup including the SNN that is connected to the sensors and SMA driver.

**Figure 5 biomimetics-09-00326-f005:**
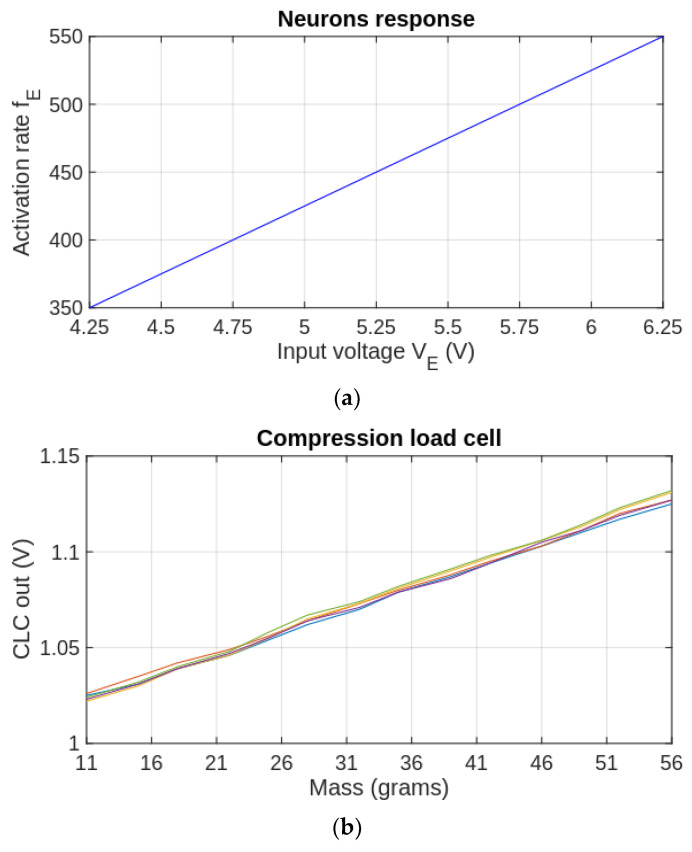
(**a**) The influence of the input voltage on the activation rate of the SOMA, and (**b**) the variation of the CLC output with the mass that is placed above the sensor; The set of measurements is repeated 5 times and the results are shown by different colours.

**Figure 6 biomimetics-09-00326-f006:**
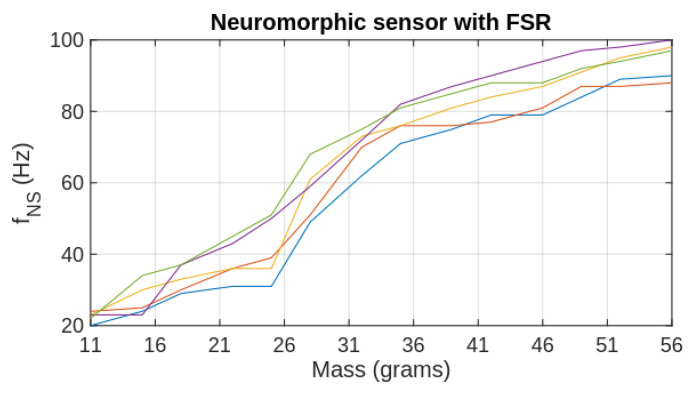
The response of the FSR-based neuromorphic sensor with the load mass; For the same mass range, we performed 5 sets of measurements that are shown with different colours.

**Figure 7 biomimetics-09-00326-f007:**
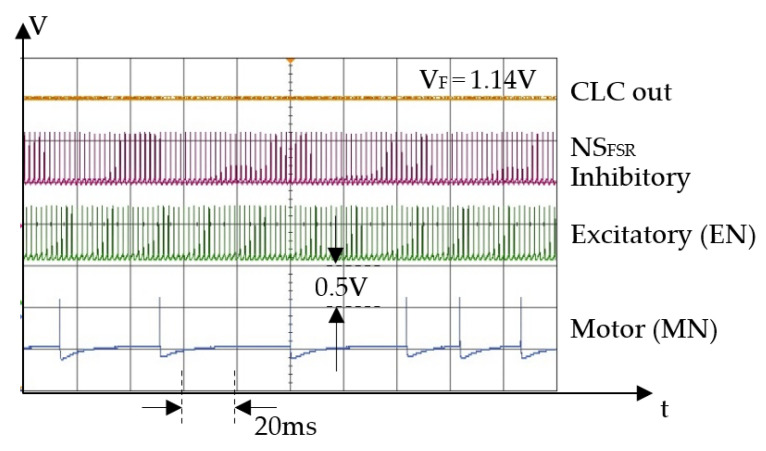
The neurons activity during steady state for different activation rates of the neurons 500 Hz when *V_E_* = 5.75 V.

**Figure 8 biomimetics-09-00326-f008:**
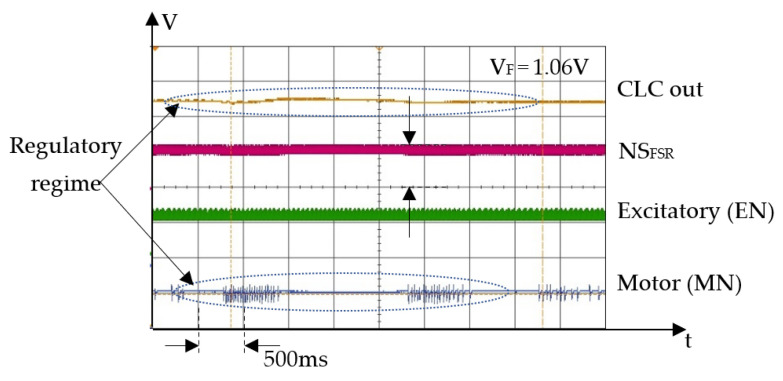
The neurons activity during steady state for different activation rates of the neurons 375 Hz when *V_E_* = 4.5 V.

**Figure 9 biomimetics-09-00326-f009:**
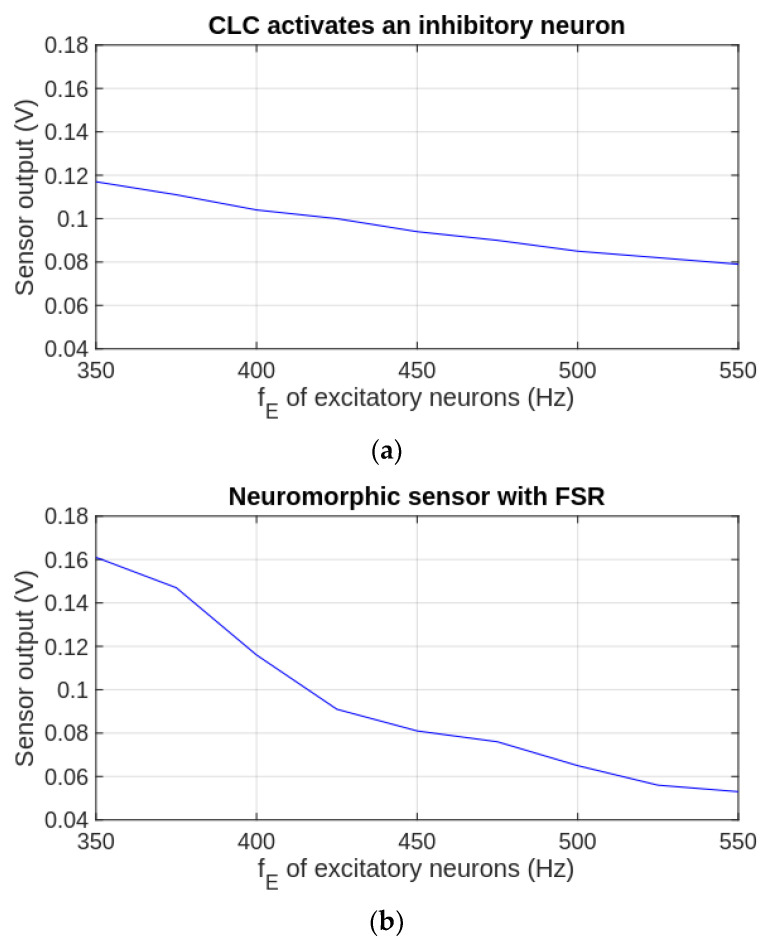
The variation of the CLC output with the spiking rate of the excitatory neurons when (**a**) the CLC activates one inhibitory neuron; (**b**) the NSFSR regulates the activity of the motor neurons.

## Data Availability

The original contributions presented in this study are included in the article; further inquiries can be directed to the corresponding authors.

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
