# Peer review of "Neuromorphic Sensor Based on Force-Sensing Resistors"

_biomimetics, 2024, doi:10.3390/biomimetics9060326_

Round 1
Reviewer 1 Report
Comments and Suggestions for Authors
1. There is no quantitative results reported in the abstract. For instance, what is the sensitivity improvement to low forces, how is the trade of lower linearity and power consumption.
2. Fig. 1 shows the circuit of the FSR, discrete components include BJT are used. However, the current approach is more toward integrated circuits (ICs) approach for higher integration, and performance. Discrete components approach is not practical. However, it would be good if the ICs could be discussed using CMOS FET.
3. Formatting: the number and unit spacing is not consistent, 24V, 115 cm, etc.
4. The experiment was carried out for the mass between 11 to 56 grams. What are the practical range of the robotics arms' masses? If the target application is for precision, what is the expected smallest manageable mass in this design? It is about the sensitivity and the range analysis.
5. Overall, a very good reference that cover the sensor to the end application.
Comments on the Quality of English Language
minor
Author Response
We would like to thank the reviewers for their constructive comments. They are very helpful for revising and improving our paper and allowed us to improve the technical contents and presentation quality. We appreciate the reviewers’ and editor’s efforts very much. We have taken into consideration and complied with all the comments and suggestions. We have studied the comments carefully and provided detailed explanations for our changes in this response. What follows are the responses to all the reviewers’ comments and suggestions.
We thank very much the distinguished reviewer for analysing carefully our work and for the very important comments that were of significant help to improve the manuscript.
Sincerely your,
Mircea Hulea,
Alexandru Barleanu

Reviewer 2 Report
Comments and Suggestions for Authors
The authors of this paper have developed a neuromorphic sensor based on electronic spiking neurons using low-cost FSRs. While the experimental design and results framework are relatively complete, there are a few points for improvement as follows:
1.When abbreviations are used for the first time, please provide the full term to enhance the readability of the article. For example, "force-sensing resistors (FSR)" is written out, but "SMA" and "SNN" are not fully spelled out upon their first appearance. Please review the entire text and improve accordingly.
2.In Figure 5(b), it is evident that the relationship between mass and voltage output is proportional. However, there are five lines of different colors in the figure, and their representations should be clearly explained. The same situation also occurs in Figure 6.
3.In Figure 5(b), it can be observed that the linear proportionality range of this study ranges from 11 to 56 grams. What potential applications of the robotic finger lie within this range? A clear description can enhance the practical value of this research.
4.The frequency 𝑓𝐸 of EN is set to several values between 350 and 550 Hz with a resolution of 25 Hz. The limitation of this interval is chosen based on empirical experience. Please provide clarification on the reference basis for these experimental experiences.
Author Response

(The authors gave the same response as above.)

Reviewer 3 Report
Comments and Suggestions for Authors
Firstly, do not use the acronyms in the title. The paper is very badly formatted, several subsections are created without subsection numbers. It is expected that a critical literature on FSR should be presented since the sensor is based on FSR. However, the authors have presented the literature with only 1 paper (ref [10]). The entire manuscript itself has 15 references, thus, indicating a missing literature review.
The motivation, problem and contributions are not presented.
The authors mentioned that the sensors proposed and "implemented in analogue hardware". However, there are no hardware details, results in the manuscript.
The methodology is not clear. How the structure in Figure 4 has been proposed.
Why linear activation function used as given in Fig 5
No comparison with other sensors in literature
For the proposed sensor, no details on framework, power consumption, calibrations are provided.
Comments on the Quality of English Language
Minor Revision
Author Response

(The authors gave the same response as above.)

Reviewer 4 Report
Comments and Suggestions for Authors
The manuscript is a research study on a neuromorphic sensor based on an electronic spiking neuron with a force-sensing resistor (FSR). The study is divided into two phases: In the first phase, the authors evaluated the response of the sensor by measuring the firing rate of the neuron (SOMA) with varying input voltage (VE) and the response of the compression load cell (CLC) and neuromorphic sensor (NSFSR) with varying load mass (mL). The results showed that the activation rate of the SOMA varied linearly with VE, and the CLC output increased linearly with mL. The NSFSR showed a logarithmic-like variation in output frequency with load mass, which is determined by the nonlinearity of the FSR's force-resistance function. In the second phase, a robotic finger actuated by shape memory alloy (SMA) was implemented. The authors evaluated the finger's ability to push and hold one tip of an elastic tweezer while the other tip was blocked. The contraction force of the actuator was adjusted by changing the input voltage (VE) that activates the excitatory neurons (EN) at a constant frequency. The activity of the motor neurons (MN), EN, and NSFSR was recorded using an oscilloscope. The results showed that the finger's force varied due to the intermittent activity of the motor neurons, with lower frequencies of EN resulting in decreased regulatory performance of the spiking neural network (SNN). The study also included the power consumption of the CLC and NSFSR. Before the final decision. There are some potential criticisms and concerns that should be addressed, including:
1. First of all, the abbreviation on the title should be expanded.
2. The manuscript included a limited number of references, and there was a lack of comprehensive literature review to support the selected problem and proposed solution methodology.
3. The literature review was not sufficient to build the motive of the study. Also, the literature review could not derive their contribution from previous studies in salient statements.
4. Neural networks play a crucial role in the proposed methodology. Specifically, the study utilized spiking neural networks (SNNs) to control the force of an anthropomorphic finger actuated by shape memory alloy (SMA). There is no justification for the use of SNN structure from the literature.
5. The manuscript provided a general overview of the experimental setup and procedures but lacked specific details on certain aspects. For example, it would be helpful to have more information on the exact configuration of the neural network, the training process, and the specific metrics used to evaluate its performance.
6. Reveising new deep learning structure is mandatorty as in: Deep Learning for Integrated Origin–Destination Estimation and Traffic Sensor Location Problems and Residual neural networks for origin–destination trip matrix estimation from traffic sensor information.
7. There is a lack of in-depth analysis and interpretation of the findings.
8. Insufficient comparison with existing approaches: The study briefly reported that the proposed neuromorphic sensor outperforms other sensing components, but there was no comparison with existing methods in the field. It is mandatory to provide a more comprehensive analysis and comparison with other relevant sensors or control systems to highlight the advantages and limitations of the proposed methodology.
9. While the study mentioned potential applications in robotics and low-power electronics, there was limited discussion on the proposed methodology's practical implications and real-world applications.
Comments on the Quality of English Language
The writing has several grammatical mistakes and inconsistencies. Some examples include inconsistent capitalization, missing articles, run-on sentences, missing or misplaced punctuation, and inconsistent verb tenses. It is essential to carefully review and edit the document to ensure clarity and accuracy.
Author Response

(The authors gave the same response as above.)

Round 2
Reviewer 2 Report
Comments and Suggestions for Authors
The author has revised the manuscript according to the previous review comments, enhancing its readability. It is recommended that this paper be accepted for publication.
Reviewer 3 Report
Comments and Suggestions for Authors
All my comments are addressed.
Comments on the Quality of English Language
Minor
Reviewer 4 Report
Comments and Suggestions for Authors
None.
Comments on the Quality of English Language
None.